# Cholesterol Exacerbates the Pathophysiology of Non-Alcoholic Steatohepatitis by Upregulating Hypoxia-Inducible Factor 1 and Modulating Microcirculatory Dysfunction

**DOI:** 10.3390/nu15245034

**Published:** 2023-12-08

**Authors:** Evelyn Nunes Goulart da Silva Pereira, Beatriz Peres de Araujo, Karine Lino Rodrigues, Raquel Rangel Silvares, Fernanda Verdini Guimarães, Carolina Souza Machado Martins, Edgar Eduardo Ilaquita Flores, Patrícia Machado Rodrigues e Silva, Anissa Daliry

**Affiliations:** 1Laboratory of Clinical and Experimental Physiopathology, Oswaldo Cruz Institute, Oswaldo Cruz Foundation, Rio de Janeiro 21040-900, RJ, Brazil; evyspereira@gmail.com (E.N.G.d.S.P.);; 2Laboratory of Inflammation, Oswaldo Cruz Institute, Oswaldo Cruz Foundation, Rio de Janeiro 21040-900, RJ, Brazil

**Keywords:** lipotoxicity, nonalcoholic fatty liver disease, microcirculation, inflammation

## Abstract

Cholesterol is a pivotal lipotoxic molecule that contributes to the progression of Non-Alcoholic Steatohepatitis NASH). Additionally, microcirculatory changes are critical components of Non-Alcoholic Fatty Liver Disease (NAFLD) pathogenesis. This study aimed to investigate the role of cholesterol as an insult that modulates microcirculatory damage in NAFLD and the underlying mechanisms. The experimental model was established in male C57BL/6 mice fed a high-fat high-carbohydrate (HFHC) diet for 39 weeks. Between weeks 31–39, 2% cholesterol was added to the HFHC diet in a subgroup of mice. Leukocyte recruitment and hepatic stellate cells (HSC) activation in microcirculation were assessed using intravital microscopy. The hepatic microvascular blood flow (HMBF) was measured using laser speckle flowmetry. High cholesterol levels exacerbated hepatomegaly, hepatic steatosis, inflammation, fibrosis, and leukocyte recruitment compared to the HFHC group. In addition, cholesterol decreased the HMBF—cholesterol-induced activation of HSC and increased HIF1A expression in the liver. Furthermore, cholesterol promoted a pro-inflammatory cytokine profile with a Th1-type immune response (IFN-γ/IL-4). These findings suggest cholesterol exacerbates NAFLD progression through microcirculatory dysfunction and HIF1A upregulation through hypoxia and inflammation. This study highlights the importance of cholesterol-induced lipotoxicity, which causes microcirculatory dysfunction associated with NAFLD pathology, thus reinforcing the potential of lipotoxicity and microcirculation as therapeutic targets for NAFLD.

## 1. Introduction

Chronic liver disease (CLD) is the leading cause of morbidity and mortality. CLD affects 844 million people, of whom 2 million die each year [1]. Non-alcoholic fatty liver disease (NAFLD) is the most prevalent CLD and is a global health challenge affecting more than a quarter of the world’s population, particularly in Western countries [2].

NAFLD is characterized by excessive fat deposits in hepatocytes in the absence of other chronic liver diseases and excessive alcohol consumption [3]. As NAFLD is associ-ated with systemic metabolic disorders, a new term has recently been proposed: metabolic dysfunction-associated steatotic liver disease [4]. This concept implies a diagnostic criteri-on directly linked to cardiometabolic risk factors without other causes of hepatic steatosis [4]. One-quarter of the affected individuals may reach an advanced pathological stage, a subtype of NAFLD histologically categorized as non-alcoholic steatohepatitis (NASH), po-tentially leading to liver fibrosis and cirrhosis, and is a significant risk factor for hepatocel-lular carcinoma [5].

Although NAFLD can be triggered by the interaction of multiple conditions as de-scribed by the multiple-hits hypothesis [6], the underlying mechanisms responsible for the progressive features of NASH are not fully understood. The disruption of hepatic choles-terol homeostasis, which leads to elevated hepatic cholesterol levels, has been linked to the development and progression of NAFLD [7], and data suggesting that free hepatic choles-terol is a major lipotoxic molecule involved in the development of both experimental and clinical NASH [8,9,10]. Hepatic lipotoxicity implies that the exposure or accumulation of toxic lipid species, such as cholesterol, in liver cells directly causes cellular toxicity, lead-ing to organelle dysfunction, chronic inflammation, and hepatocyte apoptosis, thus acting in a pro-inflammatory or pro-fibrotic manner [11,12].

One of the most critical questions is whether lipotoxicity exacerbates microcirculatory changes and contributes to NAFLD progression [13]. We previously reported that a high-fat high-carbohydrate (HFHC) diet-induced NAFLD model had an inverse association be-tween the degree of steatosis and hepatic microcirculatory blood flow [14]. Ijaz et al. demonstrated impaired hepatic microcirculation with decreased perfusion and oxygena-tion in a mouse model of cholesterol-induced moderate hepatic steatosis [15]. Moreover, our group showed the amelioration of hepatic microcirculation injury by limiting choles-terol biosynthesis using statins [14]. Thus, cholesterol metabolism may be a critical factor in the microcirculatory changes associated with NASH progression, and the main molecu-lar mechanisms involved in cholesterol-induced microcirculatory changes remain to be investigated.

In response to reduced blood flow and the consequent low-shear stress and chronic hypoxia, adaptive hypoxia-inducible factor 1 (HIF1) signaling appears to exacerbate stea-tosis, inflammation, fibrosis, and vascular dysfunction [16]. HIF1 has also been shown to play an essential role in microcirculation in various diseases such as bronchopulmonary dysplasia [17], reflux esophagitis [18], and systemic sclerosis [19]. However, research on the relationship between HIF1 signaling and microcirculatory changes associated with NASH pathogenesis is lacking. To address this question in the current study, we used in vivo approaches in a mouse model of cholesterol-associated steatohepatitis induced by an HFHC diet plus cholesterol. We investigated the metabolic, inflammatory, and molecular pathways involved in cholesterol lipotoxicity-associated effects on the microcirculation in NAFLD.

## 2. Methods

### 2.1. Animals and Experimental Procedures

In all, 39 4-week-old male C57BL/6 mice from the Central Animal Facility of the Oswaldo Cruz Foundation, Brazil, were housed in polypropylene cages in a room with a 12-h light-dark cycle and a constant temperature of 22 +/− 1 °C.

NAFLD was induced by feeding mice a long-term HFHC diet (55% Kcal fat, 35% carbohydrate) [20] with 250 g/L of fructose in drinking water (HFHC) for 31 weeks (*n* = 26). The control (CTL) group received a normocaloric diet during the same period (*n* = 13) (Nuvilab-CR1; Nuvital Nutrients Ltd., Curitiba, Brazil. Between weeks 31 and 39, a subgroup of animals on the HFHC diet received 2% cholesterol (HFHC + Col) (*n* = 13), whereas the other groups continued to receive the previous diet (Figure 1a).

The weights of each animal were measured weekly. After 39 weeks, an oral glucose tolerance test (OGTT) was conducted. At the end of the experimental protocol, the animals that had fasted overnight for 8 h were anesthetized intraperitoneally with ketamine (100 mg/kg) and xylazine (10 mg/kg) for laparotomy to expose the liver. Subsequently, microcir-culation was examined by intravital microscopy to visualize stellate cell and leukocyte re-cruitment and by laser speckle to measure basal microvascular flow [21]. After cardiac puncture for blood collection, liver and adipose tissues were collected and weighed ac-cordingly. For further analysis, serum and tissue were stored at −80 °C.

All experiments were conducted per internationally accepted principles for the care and use of laboratory animals and were approved by the Animal Welfare Committee of the Oswaldo Cruz Foundation (license L-012/2018 A1).

### 2.2. Fasting Blood Glucose

Fasting blood glucose was measured with a glucometer (Accu-Chek; Roche, South San Francisco, CA, USA) in animals that had fasted overnight for 8 h and was expressed in mmol/L.

### 2.3. Oral Glucose Tolerance Test

For the oral glucose tolerance test (OGTT), glucose at a concentration of 2 g/kg was administered orally by gavage to all animals after 6 h of fasting. Glycemia (mmol/L) was measured at the time points 0, 30, 60, 90, 120, 150, and 180 min after glucose administration with a glucometer (Accu-Chek; Roche, South San Francisco, CA, USA). The area under the curve (AUC) was determined.

### 2.4. Quantification of Insulin

Plasma insulin levels were measured in animals fasted overnight for 8 h. Analysis was performed using an enzyme-linked immunosorbent assay (Millipore, Bedford, MA, USA) kit according to the manufacturer’s guidelines.

### 2.5. Insulin Resistance Index (HOMA-IR)

HOMA-IR was calculated using the following formula: fasting insulin (mU/L) × fasting glucose (mmol/L)/22.5 [22].

### 2.6. Biochemical Parameters

Liver and serum cholesterol and triglycerides, high-density lipoprotein (HDL) and low-density lipoprotein (LDL), and alanine aminotransferase (ALT) and aspartate aminotransferase (AST) enzyme activities were determined using commercially available kits (Bioclin System II, Belo Horizonte, Brazil).

### 2.7. Histopathology

To analyze the histological features of the liver, fragments of the left lateral lobe of the liver were fixed in formalin buffered with Millonig’s phosphate buffer and then embedded in paraffin for light microscopic analysis (Nikon, model 80i and DSRi1 digital camera; Nikon Instruments, Inc., Melville, NY, USA). Hematoxylin and eosin staining was used to assess the presence of steatosis (macrovesicular, microvesicular, and total (presence of macro- or microvesicular) steatosis), inflammation, and hepatocellular damage. The samples were analyzed using a stereological grid point counting method (STEPanizer stereology version 1; Hochschulstrasse, BE, CH) [7,23]. To classify the stage of liver disease, the previously described NAFLD activity score (NAS) was used, in which the numerical scores for steatosis (0–3), hepatocellular ballooning (0–2), and lobular inflammation (0–3) were added [24]. To assess and detect the presence of fibrosis, Masson’s trichrome staining method was used to quantify collagen with ImageJ (ImageJ version 1.53e; Madison, WI, USA) [25]. For each analysis, we used five histologic sections from six mice per group.

### 2.8. Immunohistochemistry

Immunohistochemical analysis was conducted by targeting the smooth muscle alpha-actin (αSMA) to detect the activation of hepatic stellate cells [13]. To block non-specific binding, the histological sections were treated with 3% hydrogen peroxide (H_2_O_2_) after deparaffinization. Non-specific binding was blocked with skimmed milk powder and bovine serum albumin. Sections were incubated overnight in a humidified chamber at 4 °C with a primary mouse anti-αSMA monoclonal antibody (1:500, Santa Cruz Biotechnology; Santa Cruz, CA, USA). In addition, an incubation with biotinylated secondary antibody was performed, followed by an incubation with streptavidin peroxidase at room temperature. Diaminobenzidine was used as a chromogen. Finally, counterstaining with Mayer’s hematoxylin was performed.

### 2.9. Total and Differential Counts of White Blood Cells

For total and differential counts of leukocytes, blood was collected from the tails of the animals. Total cell counts were determined in a Neubauer chamber under a light microscope (40× objective) in which 5 µL of blood was diluted with 195 µL of Türk’s solution. Differential cell analysis was performed on stained blood smears using the May–Grünwald–Giemsa method [26]. Analysis was performed under a light microscope with an oil immersion objective at 100× magnification.

### 2.10. Quantification of Hepatic Myeloperoxidase

Hepatic myeloperoxidase was used to assess neutrophil recruitment. Liver tissue samples were homogenized in 1 mL of Hank’s solution. The samples were, then, subjected to osmotic shock to promote red blood cell lysis. They were homogenized in ethyldimethylethylammonium bromide (HTBA. SIGMA, St. Louis, MO, USA) to lyse neutrophils and release their granule contents. Into a 96-well plate, 50 µL each of the sample, the HTBA solution, and ortho-dianisidine (0.68 mg/mL) were added, and the plate was kept at a temperature of 37 °C for 15 min. Then, 50 µL of H_2_O_2_ (0.006%) was added to each well, and, after 10 min, 50 µL of sodium azide (1%) was added to stop the reaction. Analysis was performed by measuring absorbance using a SpectraMax M5 plate reader (Molecular Devices, San Jose, CA, USA) at a wavelength of 460 nm [27].

### 2.11. Flow Cytometry

A Cytometric Bead Array Mouse Th1/Th2/Th17 Cytokine Kit (Becton Dickinson and Company, Franklin Lakes, NJ, USA) was used for hepatic cytokine analysis. Liver tissue fragments were thawed and lysed by sonication in a phosphate buffer containing a cocktail of protease inhibitors (Roche, South San Francisco, CA, USA) and Nonidet P40 (SIGMA, St. Louis, MO, USA) at a final concentration of 1%. The samples were centrifuged at 500× *g* for 5 min, and the supernatant was removed. Subsequently, 25 µL each of the samples, the capture beads mixture, and the fluorochrome phycoerythrin were incubated for 2 h at room temperature. After incubation, the mixture was washed with 500 µL of wash buffer and centrifuged at 200× *g* for 5 min. The supernatant was discarded and resuspended in 100 µL of wash buffer. The data were acquired using a Cytoflex flow cytometer (Beckman Coulter, Indianapolis, IN, USA). The protein concentration in the liver tissue used for normalization was determined using the BCA protein assay kit (Thermo Fisher Scientific, Lenexa, KS, USA), according to the manufacturer’s recommendations.

### 2.12. RT-PCR

To perform total RNA extraction, we employed 25 µg of hepatic tissue and the RNeasy Mini Kit (Qiagen, Germantown, MD, USA). In assessing the quality and quantity of RNA, we utilized the Nanodrop spectrophotometer (Thermo Fisher, Rockford, IL, USA), with 1 µL of RNA directed towards complementary DNA (cDNA) synthesis using the High Capacity cDNA Reverse Transcription Kit (Applied Biosystems, Middletown, CT, USA). Subsequently, the cDNA was amplified in a real-time PCR reaction of 20 µL, consisting of 10 µL SYBR Green, 1.6 µL of primers (forward and reverse), 3.4 µL of distilled water, and 5 µL of diluted cDNA (1:2). In calculating the relative expression, we employed the 2^−ΔΔCt^ method, normalizing the results by beta-actin expression. The primers used were designed for the amplification of the following: collagen type I alpha 1 chain (COL1A1): forward 5′-CTCCTGGCAAGAATGGAGAT-3′ and reverse 5′-AATCCACGAGCACCCTGA-3′; HIF1A: forward 5′-GGGGAGGACGATGAACATCAA-3′ and reverse 5′-GGGTGGTTTCTTGTACCCACA-3′; intercellular adhesion molecule 1 (ICAM1): forward 5′-GTGATGGCAGCCTCTTATGT-3′ and reverse 5′-GGGCTTGTCCCTTGAGTTT-3′; nuclear factor kappa B (NF-kB): forward 5′-GAAGTGAGAGAGTGAGCGAGAGAG-3′ and reverse 5′-CGGGTGGCGAAACCTCCTC-3′; and actin beta: forward 5′-AGATTACTGCTCTGGCTCCTAGC-3′ and reverse 5′-ACTCATCGTACTCCTGCTTGCT-3′.

### 2.13. Microcirculation Parameters

#### 2.13.1. Intravital Microscopy

To assess liver microcirculation, the left lobe of the liver was externalized. The interaction between leukocytes and the endothelium was assessed by counting labeled leukocytes (0.3 mg/kg rhodamine 6G, iv) rolling or adherent in the microcirculation. Using the ultraviolet filter method, the number of vitamin A-positive hepatic stellate cells (HSCs) was determined based on endogenous vitamin A fluorescence [21].

#### 2.13.2. Blood Flow Measurement by Laser Speckle

Laser speckle contrast imaging (Pericam PSI system, Perimed, Sweden) was used to monitor hepatic microvascular blood flow expressed in arbitrary perfusion units [13].

### 2.14. Statistical Analyzes

The results are presented as the mean (standard deviation of the mean) for each group. Normal distribution was assessed using the Shapiro–Wilk test and confirmed by a QQ plot. Comparisons between groups were made using the one-way test analysis of variance (ANOVA) and Tukey’s post-hoc test for multiple comparisons to determine which group differed from the others. The relationship between the hepatic transcripts of HIF1A and basal hepatic microvascular blood flow was examined using Pearson’s correlation coefficient. The data were analyzed using GraphPad Prism 8.0.1 (GraphPad Software Inc., La Jolla, CA, USA) and R software (version 3.4.2; R Core Team, R Foundation for Statistical Computing, Wieden, Vienna, Austria). Statistical significance was set at *p* < 0.05.

## 3. Results

### 3.1. Cholesterol Affects Metabolic and Hepatic Parameters in Mice on the HFHC Diet

Compared to the mice on the CTL diet, those on the HFHC and HFHC + Col diets gained more weight (Figure 1b). Mice on the HFHC diet had an increased liver weight compared to the CTL group (Figure 1c,e), and the HFHC + Col group showed a significant increase in liver weight compared to the CTL and HFHC groups (Figure 1c,e). Importantly, the weight of the liver relative to the whole body weight (hepatosomatic index) evidenced the hepatomegaly in mice fed 2% cholesterol. Macroscopically, the HFHC and HFHC + Col groups showed pale yellow staining of the liver, indicating hepatic steatosis, a condition that was more pronounced in the mice fed 2% cholesterol (Figure 1d).

Mice fed the HFHC diet exhibited decreased glucose tolerance compared with those fed the control and HFHC + Col diets, as indicated by a higher post-challenge glucose peak and delayed return to baseline with a larger AUC (Figure 1e). In addition, mice on the HFHC diet had higher fasting blood glucose levels and insulin secretion than those on the CTL and HFHC + Col diets, with insulin resistance observed as an increase in HOMA-IR compared to the other groups (Figure 1f–h).

Mice fed the HFHC and HFHC + Col diets showed a significant increase in serum LDL cholesterol, liver triglycerides, and cholesterol compared to those in the CTL group. In addition, an increase in serum total cholesterol, HDL, and triglycerides was observed in the HFHC + Col group compared to that in the CTL group. Serum HDL, triglycerides, liver cholesterol, and triglycerides were aggravated in HFHC + Col mice compared to other groups (Figure 1i).

The HFHC and HFHC + Col groups exhibited increased ALT enzyme activity compared with the CTL group. In contrast, only the HFHC group showed increased AST enzyme activity compared with the CTL group (Figure 1j).

### 3.2. Cholesterol Worsens Histological Features of NASH

Microscopic histological analysis showed that mice fed with HFHC + Col livers had increased steatosis, inflammatory foci, and hepatocyte ballooning compared to the CTL and HFHC groups (Figure 2 and Figure 3a–f). Macrovesicular steatosis was increased in mice fed the HFHC and HFHC + Col diets compared to those fed the CTL diet (Figure 2 and Figure 3b). This was reflected in the overall liver injury assessed by the NAS score, with mice in the CTL group classified as not having NASH and those in the HFHC and HFHC + Col groups classified as having NASH; the cholesterol-supplemented group had a higher NAS score (Figure 3g).

The HFHC and HFHC + Col groups exhibited pericellular fibrosis, which occurred predominantly in lobular zone 3 and was more pronounced in the group with high cholesterol levels (Figure 2 and Figure 3h). High-cholesterol feeding increased collagen deposition compared to that in the CTL and HFHC groups, as measured by COL1A1 transcription analysis in the liver using quantitative PCR (qPCR) (Figure 3i).

### 3.3. Cholesterol Exacerbates Microcirculatory Dysfunction and Increases HIF1A

HSC activation was increased in HFHC and, in particular, HFHC + Col mice compared to that in the CTL group on a standard diet, as evidenced by the decreased number of vitamin A-positive cells in intravital microscopy analysis (Figure 4 and Figure 5a) and more excellent immunostaining for alpha-smooth muscle actin (Figure 4 and Figure 5b).

Fluxometric analysis showed that the HFHC and HFHC + Col groups had decreased basal microvascular blood flow in the liver compared to the CTL group, which was further reduced in the HFHC + Col group compared to that in the HFHC group (Figure 6 and Figure 7a). As shown by qPCR analysis of liver tissue, the HFHC + Col group showed increased mRNA transcripts for HIF1A compared to CTL and HFHC mice (Figure 7b). The hepatic transcripts for HIF1A and basal hepatic microvascular blood flow had strong negative Pearson correlation coefficients, suggesting that these variables were inversely proportional (Figure 7c).

The mice on HFHC and HFHC + Col diets had significantly higher numbers of leukocytes rolling and adhering to the hepatic microcirculation than those in the CTL group, which were exacerbated by the addition of cholesterol to the diet, as shown by the higher numbers of leukocytes rolling and adhering in the HFHC + Col group than in the HFHC group (Figure 7d,e). Accordingly, mice on the HFHC + Col diet showed an increase in mRNA transcripts for ICAM1 compared to the CTL and HFHC mice, as shown by qPCR analysis (Figure 7f).

### 3.4. Cholesterol—Increased Systemic and Hepatic Pro-Inflammatory Status

Dietary cholesterol-induced leukocytosis in peripheral blood was characterized by increased lymphocytes, monocytes, and neutrophils compared to the HFHC group (Figure 8a). A decrease in total leukocytes and neutrophils was also observed in the FHC group compared to the CTL group (Figure 8a).

Mice in the HFHC and HFHC + Col groups showed increased myeloperoxidase activity in the liver tissue compared to those in the CTL group, which was exacerbated by the addition of cholesterol to the diet, as demonstrated by higher myeloperoxidase activity in the HFHC + Col group than in the HFHC group (Figure 8b).

The addition of 2% cholesterol increased mRNA transcripts for NF-kB compared with the CTL and HFHC groups, as shown by qPCR analysis in liver tissue (Figure 8c).

Significant increases in Th1 (IL -2, IL -6, IFN-γ, and TNF-α) (Figure 8d), Th2 (IL-4 and IL-10) (Figure 8e), and Th17 (IL-17) (Figure 8f) cytokines and chemokines were observed in the livers of the mice from the HFHC + Col group compared with those in the CTL and HFHC groups. The IFN-γ/IL -4 ratio, indicative of a Th1-type pattern, was higher in mice fed cholesterol than those on a CTL diet. This evidence of polarization was not observed in the HFHC group (Figure 8g).

## 4. Discussion

In the present study, we addressed the role of cholesterol as an aggravating factor in microcirculatory disturbances associated with NAFLD progression. Although cholesterol is a cytotoxic lipid [28], its role and mechanisms of action in NAFLD progression are not fully understood.

We demonstrated that cholesterol exacerbates the characteristics of NASH, such as increased serum and liver lipid profiles, steatosis, fibrosis, and inflammation. Moreover, a cholesterol-supplemented diet leads to microcirculatory damage and increased HIF1A levels, strengthening lipid metabolism and an important target for NAFLD therapeutics.

The exacerbation of serum and liver cholesterol and triglyceride levels, culminating in increased steatosis, hepatomegaly, fibrosis, and inflammation, was reflected by an increase in the NAS of mice supplemented with cholesterol. These phenomena are associated with an imbalance between the synthesis and release (excessive lipolysis or excessive synthesis due to obesity), storage (plasma absorption and de novo lipogenesis), and catabolism of free fatty acids (β-oxidation and plasma secretion), as very high LDL is characteristic of NAFLD [8,29,30]. The evidence suggests that overloading the liver with free cholesterol leads to lipotoxicity and is a mechanistic factor that exacerbates inflammation and fibrosis [31], contributing to the development and progression of NAFLD.

In the microcirculation, we demonstrated that the decrease in blood flow was enhanced by high levels of dietary cholesterol, which could be explained by (a) compression of the sinusoids due to steatosis and liver fibrosis, (b) activation and increased contractility of HSCs due to lipotoxicity and inflammation, and (c) increased leukocyte recruitment, which might exacerbate sinusoid compression.

Lipotoxicity contributes to the amplification and maintenance of the inflammatory response. In this regard, we observed that the cholesterol-supplemented mice exhibited increased numbers of circulating neutrophils, leukocytes, and monocytes and, in the liver, increased myeloperoxidase activity and release of cytokines and chemokines such as IL-2, IL-17, IL-6, IFN-γ, TNF-α, IL-4, and IL-10. The cholesterol-supplemented mice also showed a polarization toward a Th1 response, as shown by the IFN-γ/IL-4 ratio increase. The accumulation of toxic lipids can polarize Th1 and Th17 responses to NASH [32]. Th1 cells can promote the differentiation of macrophages into M1 macrophages, which play a pro-inflammatory role by secreting IFN-γ and TNF-α [33,34]. Evidence suggests that a Th1 response may be detrimental, and a Th2 response is beneficial in NASH [35] because Th2 cytokines are required to inhibit the activity of pro-inflammatory Th1 cytokines. However, as shown previously [36,37], the Th2 immune response is critical for tissue repair after injury. However, when this process becomes chronic, hyperactive, or dysregulated, hepatocyte expansion and extracellular matrix remodeling lead to the restoration of the lost population and pathological fibrosis [36,37], respectively, as shown in our study. The Th17 pattern may also enhance the fibrogenic processes by activating HSC. IL-17A directly stimulates type 1 collagen production or via autocrine IL-6 secretion in activated HSC [38,39]. Overall, IL-17A is a critical mediator of chronic liver injury, mediating the interaction between liver macrophages/Kupffer cells, HSCs, and steatotic hepatocytes and initiating the transition from simple steatosis to NASH [40,41].

Reduced blood flow to the liver, exacerbated by cholesterol, has several deleterious effects. Recent studies have shown that chronic hypoxia, leading to oxygen and nutrient deprivation, is an essential factor that exacerbates the toxic effects of lipids on hepatocytes [42]. Hypoxia triggers oxidative stress and liver inflammation, which accelerates the development of NASH and liver fibrosis. In this scenario, the inflamed steatotic hepatocytes produce more extracellular vesicles that activate HSCs and induce Kupffer cells to develop inflammatory phenotypes [42]. Steatotic hepatocytes are highly susceptible to hypoxia [43]. HIFs regulate the transcription of hypoxia-signaling pathways at the molecular level. While HIF1 beta (HIF1B) is constitutively expressed in chronic hypoxia, the α-subunit is regulated in an oxygen-dependent manner. Increased HIF1 alpha (HIF1A)—a nuclear translocation factor of the aryl hydrocarbon receptor—forms a functional complex with β-subunits and transactivates more than 60 genes related to angiogenesis, inflammation, fibrosis, apoptosis, and cell proliferation [44,45].

We observed increased HIF1A expression, concomitant with ICAM-1, and pro-inflammatory and tissue repair cytokine expression in the livers of cholesterol-induced NASH mice. In hypoxia situations, the increase in HIF1 leads to increased activation of HSCs by decreasing hydroxylation and degradation of HIF1A. However, maximal activation of HIF1 requires the activation of other signaling pathways, such as NF-kB, which is activated in the first hours of chronic hypoxia [46]. This leads to an increase in HIF1A transcription due to the presence of an NF-kB binding site in the promoter of the HIF1A gene [47,48,49]. NF-kB activation, therefore, leads to increased expression of HIF1A mRNA [50]. The induction of HIF1A expression positively regulates inflammatory signaling and increases the expression of adhesion molecules.

Activation of HIF1A may also explain the normalization of fasting blood glucose and insulin levels in mice with NASH supplemented with cholesterol. In concordance, Eng et al. showed that mice fed a high-fat/high-fructose diet supplemented with 2% cholesterol showed increased liver inflammation and fibrosis. In contrast, insulin resistance and glucose intolerance were less pronounced than in mice fed a similar diet without cholesterol [51]. Previous data show that Glucose transporter 1 (GLUT1) is upregulated under hypoxic conditions by the interaction of HIF-1A with a hypoxia-responsive element site of the GLUT1 promoter, as has been described for several other genes regulated by low oxygen availability. This phenomenon reflects the adaptation of glucose metabolism to hypoxic conditions [52]. This increase in GLUT1 expression is accompanied by increased glucose uptake and glycolysis.

### Limitations

Blood glucose measurements using a glucometer (Accu-Chek; Roche, South San Francisco, CA, USA) deserved attention. Although glucometers have advantages such as low consumption of blood samples and cost-effectiveness of repeated measurements, the precision and accuracy of these devices are questionable and controversial in the literature [53]. To mitigate these disadvantages, we attempted to reduce these biases. Thus, measurements were conducted using the same device under similar conditions, and the same expert conducted all examinations.

Non-alcoholic fatty liver disease (NAFLD) is a multifactorial pathogenetic phenomenon. An imbalance in lipid homeostasis (synthesis, uptake, and metabolism) is a critical factor in liver disease. Nevertheless, several new pathogenic phenomena have been discussed [54,55,56]. This study demonstrated that upregulation of HIF1A and microcirculatory dysfunction are involved in the progression of NAFLD by cholesterol administration. However, cholesterol may also exacerbate NAFLD progression through other mechanisms. Further studies are needed to support the hypothesis that cholesterol drives microcirculatory changes and HIF1A upregulation during NAFLD progression.

## 5. Conclusions

Based on our findings, we concluded that cholesterol plays a substantial role in NAFLD progression and its associated complications. This study elucidated the effects of high dietary cholesterol levels on NAFLD, which led to increased serum and liver lipid profiles, steatosis, hepatomegaly, fibrosis, and inflammation. Our research highlights a new aspect of cholesterol toxicity: its negative effects on microcirculation, leading to decreased blood flow in the liver. This reduced blood flow can have detrimental consequences for hepatocytes, aggravating the toxic effects of lipids and consequently causing inflammation. Remarkably, we demonstrated for the first time in vivo that high levels of dietary cholesterol lead to increased activation of HSCs and recruitment of leukocytes into the hepatic microcirculation. Furthermore, we showed unprecedentedly that cholesterol-induced NAFLD was associated with increased expression of HIF1A, a critical factor that regulates several genes related to angiogenesis, inflammation, fibrosis, apoptosis, and cell proliferation. Interestingly, our research also suggests cholesterol can affect glucose metabolism and potentially improve glucose metabolism in mice with NAFLD, possibly through mediated upregulation of HIF1A.

In summary, this study highlighted the role of cholesterol in exacerbating NAFLD progression by contributing to liver injury, inflammation, and microcirculatory dysfunction. A comprehensive understanding of the mechanisms underlying cholesterol-triggered NAFLD progression, including the involvement of HIF1A and its effects on microcirculation and inflammation, may have important implications in developing therapies targeting this condition (Figure 9).

## Figures and Tables

**Figure 1 nutrients-15-05034-f001:**
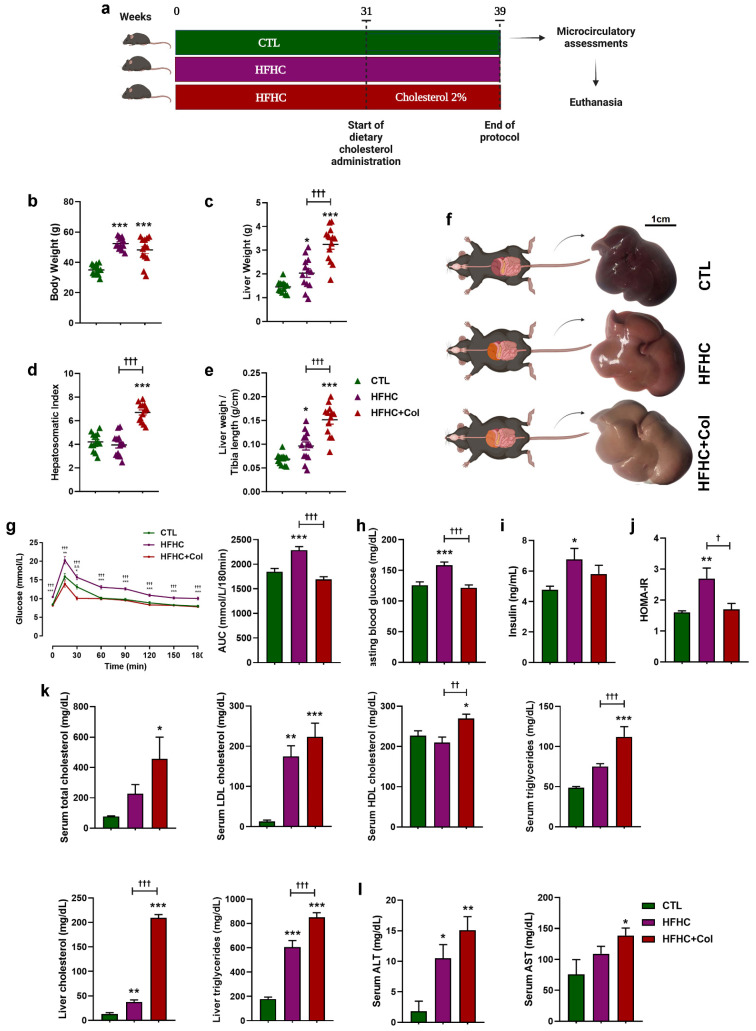
Cholesterol alters the hepatic and metabolic parameters in mice with NAFLD. (**a**) Schematic representation of the experimental procedure used in the present study to determine the influence of dietary cholesterol on the progression of NAFLD. (**b**) Liver and metabolic parameters of mice fed a control (CTL), high-fat, high-carbohydrate (HFHC), and HFHC plus 2% cholesterol (HFHC + Col) diet are shown as follows: (**b**) body weight, (**c**) liver weight, (**d**) hepatosomatic index, (**e**) liver weight/tibial length, (**f**) macroscopic examination of the liver (scale bar: 1 cm), (**g**) assessment of glucose metabolism by oral glucose tolerance test with calculation of area under the curve according to the trapezoidal rule, (**h**) fasting blood glucose, (**i**) insulin, (**j**) HOMA-IR, and (**k**,**l**) biochemical parameters. Data are expressed as the mean ± SEM with * *p* < 0.05 vs. CTL; ** *p* < 0.01 vs. CTL; *** *p* < 0.001 vs. CTL; ^† ^*p* < 0.05 vs. HFHC; ^†† ^*p* < 0.01 vs. HFHC; ^††† ^*p* < 0.001 vs. HFHC and ^&& ^*p* < 0.001 vs. HFHC + Col, for the one-way analysis of variance (ANOVA) with Tukey’s multiple comparisons post-hoc test.

**Figure 2 nutrients-15-05034-f002:**
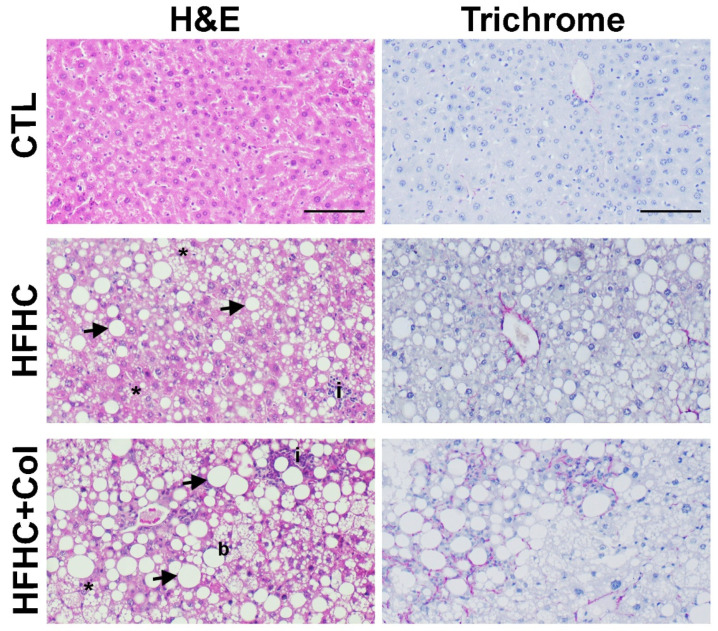
Cholesterol exacerbates features of NASH. Representative images of histological analysis with hematoxylin and eosin staining and Masson’s trichrome staining of livers from mice that are fed a control (CTL), high-fat, high-carbohydrate (HFHC), and HFHC plus 2% cholesterol (HFHC + Col) diet are shown (scale bars: 50 μm). Arrow: macrovesicular steatosis; asterisk: microvesicular steatosis; letter b: hydropic degeneration; letter i: inflammatory foci.

**Figure 3 nutrients-15-05034-f003:**
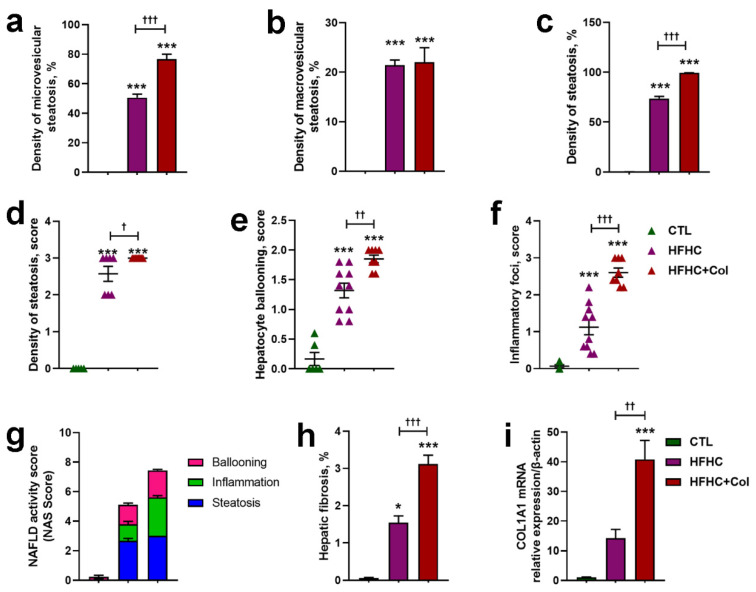
Cholesterol exacerbates features of NASH. Quantification of the percentage of hepatic macrovesicular, microvesicular, and total steatosis is shown in (**a**–**c**), (**d**) steatosis, (**e**) hepatocyte ballooning, (**f**) inflammatory foci scores, and (**g**) NAFLD activity score. In (**h**), the quantification of the percentage of liver fibrosis area is depicted. Transcription of the gene encoding collagen type I alpha 1 was examined using real-time PCR, as shown in (**i**). Data are expressed as the mean ± SEM with * *p* < 0.05 vs. CTL; *** *p* < 0.001 vs. CTL; ^† ^*p* < 0.05 vs. HFHC; ^†† ^*p* < 0.01 vs. HFHC; and ^††† ^*p* < 0.001 vs. HFHC, for the one-way analysis of variance (ANOVA) with Tukey’s multiple comparisons post-hoc test.

**Figure 4 nutrients-15-05034-f004:**
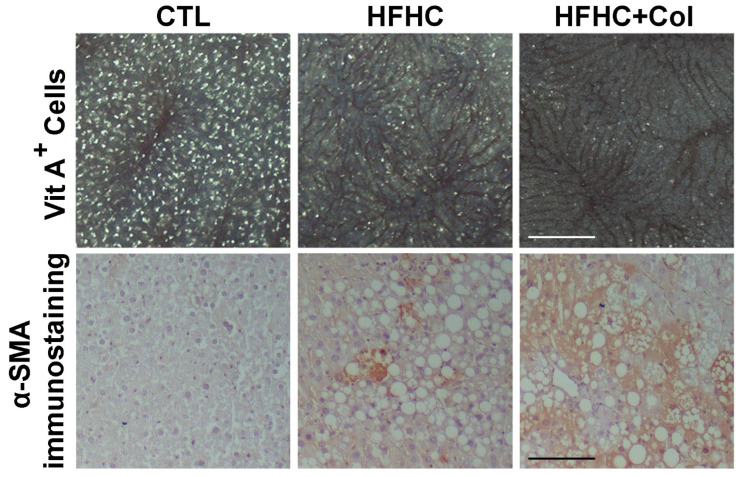
Cholesterol exacerbates microcirculatory injury and increases hypoxia-inducible factor 1 alpha in the liver of mice with NASH. The microcirculation was assessed in mice that were fed a control (CTL), high-fat, high-carbohydrate (HFHC), and HFHC plus 2% cholesterol (HFHC + Col) diet. Intravital microscopy was used to examine the distribution of vitamin A-positive cells in vivo, indicating the storage of retinoids in cytoplasmic droplets of hepatic stellate cells. The percentage of the SMA-positive area is demonstrated (scale bars: 50 μm).

**Figure 5 nutrients-15-05034-f005:**
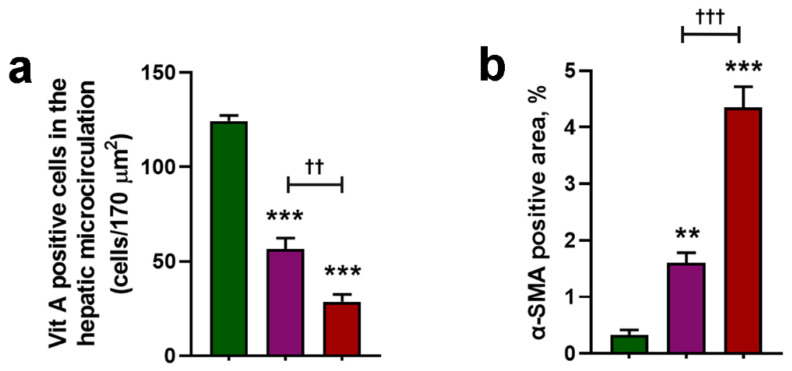
Cholesterol exacerbates microcirculatory injury and increases hypoxia-inducible factor 1 alpha in the liver of mice with NASH. The microcirculation was assessed in mice that were fed a control (CTL), high-fat, high-carbohydrate (HFHC), and HFHC plus 2% cholesterol (HFHC + Col) diet. Intravital microscopy was used to examine the quantification (**a**) of vitamin A-positive cells in vivo, indicating the storage of retinoids in cytoplasmic droplets of hepatic stellate cells. In (**b**), the percentage of the SMA-positive area is quantified. Data are expressed as the mean ± SEM with ** *p* < 0.01 vs. CTL; *** *p* < 0.001 vs. CTL; ^†† ^*p* < 0.01 vs. HFHC; and ^††† ^*p* < 0.001 vs. HFHC, for the one-way analysis of variance (ANOVA) with Tukey’s multiple comparisons post-hoc test.

**Figure 6 nutrients-15-05034-f006:**
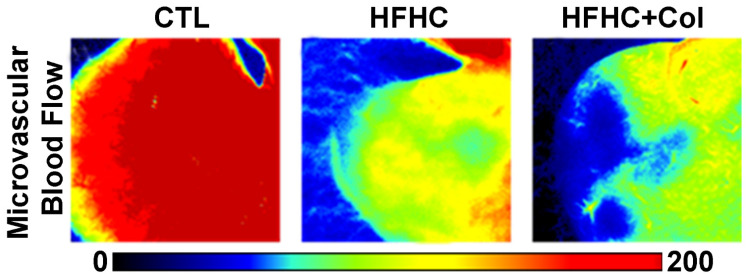
Cholesterol exacerbates microcirculatory injury and increases hypoxia-inducible factor 1 alpha in the liver of mice with NASH. Microcirculation was assessed in mice that were fed a control (CTL), high-fat, high-carbohydrate (HFHC), and HFHC plus 2% cholesterol (HFHC + Col) diet. Liver microvascular blood flow was examined using laser speckle contrast imaging (scale bars: 50 μm).

**Figure 7 nutrients-15-05034-f007:**
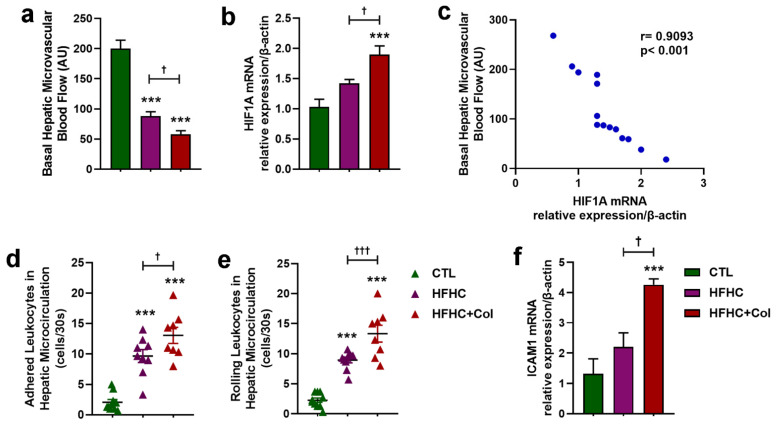
Cholesterol exacerbates microcirculatory injury and increases hypoxia-inducible factor 1 alpha in the liver of mice with NASH. The microcirculation was assessed in mice that were fed a control (CTL), high-fat, high-carbohydrate (HFHC), and HFHC plus 2% cholesterol (HFHC + Col) diet. Quantification of microvascular blood flow in the left lateral lobe of the liver is shown in (**a**). The color scale indicates blood flow in arbitrary perfusion units. Real-time PCR analysis of the mRNA transcript levels of the gene encoding hypoxia-inducible factor 1 alpha (HIF1A) is shown in (**b**). Pearson correlation analysis between hepatic transcripts for HIF1A and basal hepatic microvascular blood flow is shown in (**c**). After intravenous administration of rhodamine 6G, leukocyte recruitment was examined using intravital microscopy. Quantification of (**d**) leukocyte adhesion and (**e**) rolling is shown. The mRNA transcript levels of the gene encoding intercellular adhesion molecule 1 are shown in (**f**). Data are expressed as the mean ± SEM with *** *p* < 0.001 vs. CTL; ^† ^*p* < 0.05 vs. HFHC; and ^††† ^*p* < 0.001 vs. HFHC, for the one-way analysis of variance (ANOVA) with Tukey’s multiple comparisons post-hoc test.

**Figure 8 nutrients-15-05034-f008:**
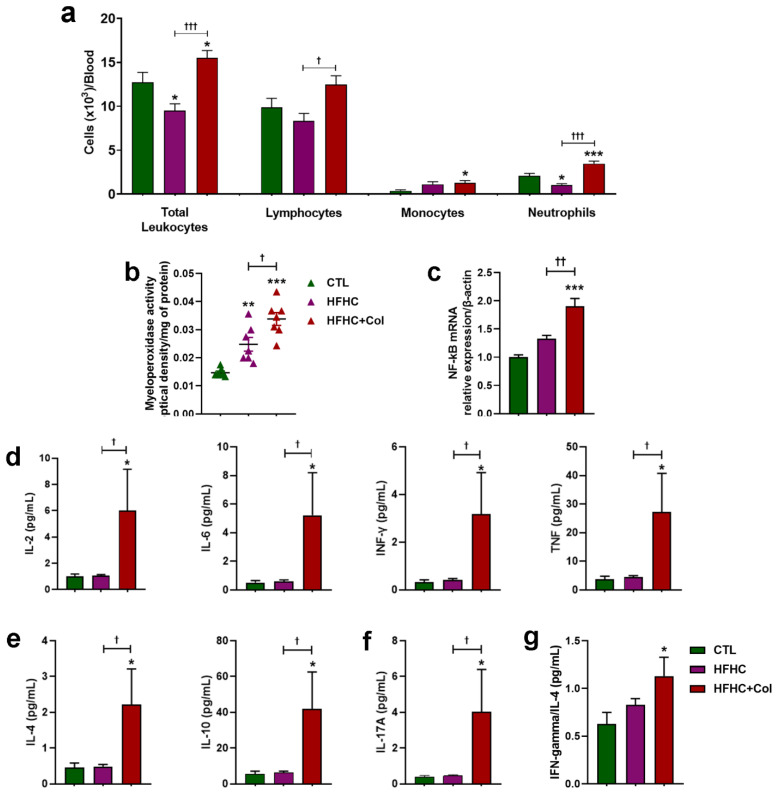
Cholesterol alters the inflammatory profile of the blood and liver of mice with NASH. Analyses of animals being fed control (CTL), high-fat, high-carbohydrate (HFHC), and HFHC plus 2% cholesterol (HFHC + Col) diets were performed in the peripheral blood by (**a**) total and differential leukocyte counts, and in liver tissue by (**b**) quantification of myeloperoxidase activity, (**c**) evaluation of mRNA transcript levels of the gene encoding nuclear factor kappa B, and quantification of (**d**) Th1, (**e**) Th2, and (**f**) Th17 cytokines and chemokines levels using flow cytometry. The IFN-γ/IL-4 ratio (indicative of a Th-1 pattern) is shown in (**g**). Data are expressed as the mean ± SEM with * *p* < 0.05 vs. CTL; ** *p* < 0.01 vs. CTL; *** *p* < 0.001 vs. CTL; ^† ^*p* < 0.05 vs. HFHC; ^†† ^*p* < 0.01 vs. HFHC; and ^††† ^*p* < 0.001 vs. HFHC, for the one-way analysis of variance (ANOVA) with Tukey’s multiple comparisons post-hoc test.

**Figure 9 nutrients-15-05034-f009:**
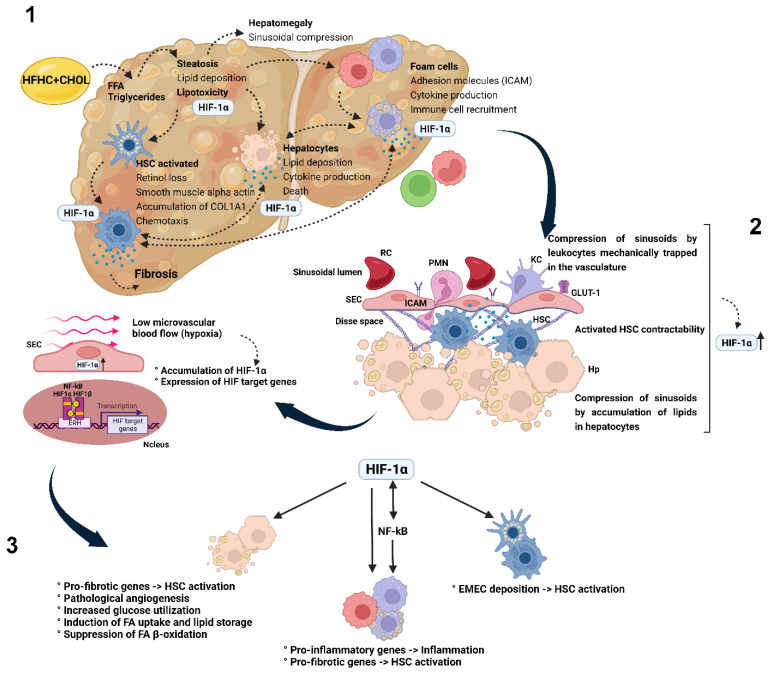
Effect of cholesterol as a lipotoxic agent that aggravates microcirculatory damage in NASH. (1) Consumption of a high-fat, high-carbohydrate diet, which is associated with an increase in cholesterol levels, leads to an increase in circulating fatty acids. Excess fatty acids and triglycerides lead to hepatic steatosis. In this context, the disturbance of lipid metabolism and storage leads to lipotoxicity, the first insult of cells, especially in hepatocytes. Lipotoxicity can also affect Kupffer and hepatic stellate cells (HSCs), and all three cells can release a myriad of interleukins and chemokines in response to their activation. When Kupffer cells take up large amounts of oxidized lipids, they form foam cells. The foam cells, then, synthesize interleukins, which stimulate the recruitment of more inflammatory cells, triggering a vicious cycle of inflammation. The stellate cells switch from a quiescent to an activated profibrotic phenotype. HSC activation is characterized by increased collagen production. Chronic damage, then, exceeds the ability of the HSCs to restore functional tissue, and fibrosis and intercellular scarring develop. (**2**) Consequently, dietary cholesterol acts on various damages to the microcirculation, which together contribute to the progression of NAFLD to NASH. In this context, the basal blood flow decreases in the hepatic microcirculation resulting from an increase in lipid deposition, with a marked increase in HSC activation, from quiescent cells to cells that take on myofibroblastic properties, leading to increased collagen deposition and leukocyte recruitment in sinusoidal and postsinusoidal venules. (**3**) HIF1A transcription is increased when the blood supply to the hepatic microvasculature is reduced. Low shear stress stimulates nuclear factor kappa B (NF-kB), inducing HIF1A. Impaired blood flow is also capable of inducing NF-kB-independent HIF1A expression mediated by reactive oxygen species. Induction of HIF1A expression leads to the upregulation of inflammatory signaling via the activation of NF-kB and expression of adhesion molecules, including ICAM-1. Along with positive inflammatory regulation, there is an increase in glycolysis, with binding of HIF1A to hypoxia response elements in its promoters directly responsible for the increased expression of glycolytic enzymes and glucose transporters, with one target gene being a uniport glucose transporter.

## Data Availability

The datasets used and/or analyzed in the current study are available from the corresponding author upon reasonable request.

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
