# Peer review of "Cholesterol Exacerbates the Pathophysiology of Non-Alcoholic Steatohepatitis by Upregulating Hypoxia-Inducible Factor 1 and Modulating Microcirculatory Dysfunction"

_nutrients, 2023, doi:10.3390/nu15245034_

Round 1
Reviewer 1 Report
Comments and Suggestions for Authors
Overall Assessment:
The article, titled "CHOLESTEROL DRIVES MICROCIRCULATORY CHANGES THROUGH HYPOXIA-INDUCIBLE FACTOR 1 UPREGULATION DIFFERENTIALLY MODULATING THE PROGRESSION TO NONALCOHOLIC STEATOHEPATITIS," presents a comprehensive investigation into the intricate mechanisms of cholesterol-induced liver damage and its implications for nonalcoholic steatohepatitis (NASH). The study provides valuable insights into the multifaceted relationships between cholesterol, hepatic health, and NASH development.
Strengths and Contributions:
The strengths of the study lie in its in vivo experimental design, which encompasses a wide range of measurements, including insulin resistance, hepatomegaly, hepatic steatosis, inflammation, fibrosis, and microcirculatory blood flow. Additionally, the exploration of hypoxia-inducible factor 1 (HIF1A) upregulation and the immune response dynamics induced by cholesterol is both novel and intriguing. These findings make a valuable contribution to the field of NAFLD and NASH research.
Considerations and Suggestions:
However, there are some important considerations that must be addressed:
- Statistical Power and Sample Size: The manuscript should provide a thorough discussion of the statistical power in relation to the number of mice used in the study. Ensuring that the sample size was adequate to detect meaningful differences in all relevant outcomes is crucial.
- Figure Interpretability: The clarity and resolution of the figures, particularly the histological images, need improvement to enhance the interpretability of the data. The dimensions and resolution of the histological images should be optimized, accompanied by clear and informative figure legends.
- Glucose Assessment: The use of a glucometer for glucose assessment raises concerns about accuracy and precision. A discussion of the potential limitations associated with this method and consideration of more reliable laboratory-based measurements is essential.
- It is important to acknowledge that the role of cholesterol in NAFLD has been extensively studied in the existing literature. Therefore, it is advisable to reevaluate the tone and wording of certain statements in the manuscript, toning down extreme assertions and emphasizing the contribution of this study within the context of the already established knowledge base.
Addressing these concerns will enhance the manuscript's quality, accuracy, and overall impact. The article offers valuable contributions to the field of NAFLD and NASH research, and with the suggested revisions, it has the potential to significantly advance our understanding of the role of cholesterol in hepatic health and NASH progression.
In conclusion, this article is a noteworthy addition to the literature, providing insights into the intricate interplay of cholesterol-induced lipotoxicity, hepatic health, and the development of NASH. The proposed revisions will serve to further strengthen the manuscript's scientific rigor and enhance its value to the research community.
Other issues are reported in the attached annotated PDF.

Reviewer 2 Report
Comments and Suggestions for Authors
The author attempts to reveal the mechanism of action of Cholesterol Drives Microcirculatory Changes, which may be achieved by Hypoxia-Inducible Factor 1 Upregulation regulating the Progression to Nonalcoholic Steatohepatitis. The topic is of great significance and the results are rich, but the logic needs to be improved.
Research results indicate that high cholesterol levels can increase NASH symptoms such as hepatic steatosis and fibrosis. Cholesterol can reduce liver microvascular blood flow. Cholesterol can induce the activation of hepatic stellate cells and aggravate fibrosis. At this time, the expression of hypoxia-inducible factor HIF1A increases, and the IFN-γ/IL-4 ratio increases. But my biggest concern is: these independent results can only show that they are relevant in the development of NASH. The author cannot yet prove whether there is a causal relationship between them based on the existing results. It is also possible that cholesterol exacerbates the progression of NAFLD through other pathways. Microcirculatory dysfunction and HIF1A upregulation caused by hypoxia and inflammation may be a concomitant phenomenon in the development of NASH. Whether cholesterol-induced lipotoxicity and microcirculatory changes are the main factors determining NASH needs further proof, and is at least speculation.
Reviewer 3 Report
Comments and Suggestions for Authors
This article mainly investigates the mechanism of cholesterol-induced liver injury in a nonalcoholic fatty liver disease (NAFLD) model exacerbated by cholesterol feeding. As a cytotoxic lipid, cholesterol exacerbates microcirculation, inflammation, and liver fibrosis in NFALD. This study reinforces adipotoxicity and microcirculation as potential therapeutic targets for NAFLD treatment. There are still some issues in this article that need to be revised.
1. In Figure 3a, there is no scale bar.
2. In Figure 4C, the author detected NF-κB mRNA levels, however, the phosphorylated NF-κB protein regulates the expression of pro-inflammatory genes, please provide the total and phosphorylated protein results of NF-κB. Note that the abbreviation for NFKB should be NF-κB.
3. The manuscript has some formatting errors, such as line 23 IL4.
4. The author used cholesterol to exacerbate the NASH model and should add another group of normobaric diets to 31 weeks and a diet with 2% cholesterol from 31 to 39 weeks.
5. Figure 2A, please describe what the arrow, *, i and b represent.
Round 2
Reviewer 1 Report
Comments and Suggestions for Authors
The revised manuscript demonstrates commendable improvements made by the authors in addressing the prior review's criticisms. However, there remains an opportunity to further enhance the readability of the figures and graphs. Although some figures, such as 2a, have shown improvements, figures 3a and 3d continue to be presented at a reduced size. Additionally, the graphs appear overcrowded and excessively small, which may impede immediate interpretation. It's crucial for figures and graphs to succinctly present results for intuitive understanding; otherwise, they risk being counterproductive and causing disruption in reader comprehension. I strongly recommend that the authors reconsider the presentation of these figures to effectively convey their intended information without causing confusion. Furthermore, I propose separating the macro-microphotographs and graphs into distinct figures. By presenting photos and graphs in separate figures, the authors can notably enhance overall clarity and visual interpretation of their findings. This approach will provide a more focused and less crowded representation of data, ensuring each figure serves its purpose effectively. The segregation of visual elements into distinct figures will significantly improve reader comprehension and facilitate a clearer understanding of the information presented. I encourage the authors to seriously consider this suggestion, as it has the potential to substantially elevate the overall presentation of their results.
Author Response
Dear reviewer, thank you very much for revising our manuscript. According to your suggestion, we have improved the figures and the corresponding quotations in the text. Thank you very much for your valuable comments.
Reviewer 3 Report
Comments and Suggestions for Authors
No more comments
Author Response
Dear reviewer
Thank you for reviewing our manuscript.